# Turning Tertiary Lymphoid Structures (TLS) into Hot Spots: Values of TLS in Gastrointestinal Tumors

**DOI:** 10.3390/cancers15020367

**Published:** 2023-01-05

**Authors:** Daming Cai, Heng Yu, Xingzhou Wang, Yonghuan Mao, Mengjie Liang, Xiaofeng Lu, Xiaofei Shen, Wenxian Guan

**Affiliations:** 1Department of General Surgery, Affiliated Drum Tower Hospital, Medical School of Nanjing University, Nanjing 210008, China; 2Department of General Surgery, Drum Tower Clinical Medical College of Nanjing Medical University, Nanjing 210008, China

**Keywords:** tertiary lymphoid structure, gastrointestinal tumors, immune microenvironment, therapeutic targets, prediction

## Abstract

**Simple Summary:**

Despite considerable improvements in immunotherapy of cancers, how to predict and enhance the anti-tumor immune response remains unknown. As a new emerging player identified in tumor immunology, tertiary lymphoid structure (TLS) has been shown to play critical roles in local tumor immune microenvironments. TLS serves as a guardian located nearest to tumors by providing an arena for lymphocyte maturation and anti-tumor immune responses. Accumulating evidence has shown numerous aspects of TLS, which not only includes its cellular composition and formation process, but also includes how it exerts local anti-tumor immune responses. Our review summarizes recent findings in the characteristics of TLS in gastrointestinal tumors and also gives a brief introduction on how to manipulate TLS formation and anti-tumor immune responses to benefit tumor treatment in the future.

**Abstract:**

Tertiary lymphoid structures (TLSs) are ectopic lymphocyte aggregation structures found in the tumor microenvironment (TME). Emerging evidence shows that TLSs are significantly correlated with the progression of gastrointestinal tumors, patients’ prognosis, and the efficacy of adjuvant therapy. Besides, there are still some immunosuppressive factors in the TLSs that may affect the anti-tumor responses of TLSs, including negative regulators of anti-tumor immune responses, the immune checkpoint molecules, and inappropriate tumor metabolism. Therefore, a more comprehensive understanding of TLSs’ responses in gastrointestinal tumors is essential to fully understand how TLSs can fully exert their anti-tumor responses. In addition, targeting TLSs with immune checkpoint inhibitors and vaccines to establish mature TLSs is currently being developed to reprogram the TME, further benefiting cancer immunotherapies. This review summarizes recent findings on the formation of TLSs, the mechanisms of their anti-tumor immune responses, and the association between therapeutic strategies and TLSs, providing a novel perspective on tumor-associated TLSs in gastrointestinal tumors.

## 1. Introduction

The tertiary lymphoid structures (TLSs) are ectopic agminated lymphoid structures formed in sites that are normally devoid of canonical lymphoid organs. They are similar to the secondary lymphoid organs (SLOs), such as lymph nodes, the spleen, tonsils, Peyer’s patches, and mucosa-associated lymphoid tissues. The aggregate lymphoid structures are mainly formed by B cells, T cells, dendritic cells (DCs), follicular dendritic cells (FDCs), and high endothelial venules (HEVs). TLSs are most commonly found in the regions associated with chronic inflammation, infectious diseases, autoimmune diseases, transplanted organs, and tumor sites [1,2,3,4,5]. Emerging evidence has shown the importance of TLSs in anti-tumor immune responses. Recent studies reported the association of TLSs in cancer lesions with improved prognosis in several human malignancies, suggesting that the TLSs may play an important role in the antitumor immune microenvironment [6,7]. In recent years, gastrointestinal tumors posed a serious burden on people around the world due to their high morbidity and mortality. In 2018, there were an estimated 4.8 million new cases of gastrointestinal malignancy and 3.4 million disease-related deaths worldwide. Gastrointestinal tumors account for 26% of global cancer incidence, and 35% of all cancer-related deaths [8]. The presence of TLSs has been observed in gastrointestinal tumors, such as gastric cancer [9], colorectal cancer [10], and gastrointestinal stromal tumors (GISTs) [11]. Emerging evidence has also shown that TLSs were significantly correlated with the progression of gastrointestinal tumors, patients’ prognosis, and the efficacy of adjuvant therapy. Therefore, TLSs may be crucial in the gastrointestinal tumor microenvironment. It is important to deeply understand the molecular processes that lead to TLS formation in gastrointestinal tumors, the types of gastrointestinal tumors-associated TLSs, and the consequences of their presence for the generation or maintenance of tumor-specific immunity, which may benefit the development of therapeutic strategies targeting TLSs and also predicting the prognosis of patients with gastrointestinal tumors based on TLSs.

## 2. Process of Tumor-Related TLSs Formation and Maturation

The formation of SLOs, such as the spleen, lymph nodes, tonsils, Peyer’s patches, and mucosa-associated lymphoid tissue, are formed by the interaction of hematopoietic precursor lymphoid tissue-induced cells (LTi) and stromal-forming cells. The formation of TLSs is similar to SLOs, where certain pathological conditions, such as antigen-presenting cells (APCs), recognize tumor-exposed antigens and then present them to adaptive immune cells, causing activation of immune cells and secretion of cytokines [12]. These activated lymphocytes and DCs express lymphotoxin-α (LT-α), enabling them to interact with the corresponding receptor (lymphotoxin-β receptor LT-βR) expressed on stromal cells (i.e., endothelial cells, fibroblasts, and epithelial cells). LT-α can induce the expression of a variety of chemokines, including the chemokine (C-C motif) ligand 19 (CCL19), CCL21, the chemokine (C-X-C motif) ligand 12 (CXCL12), and CXCL13 [13]. The presence of CXCL13 and interleukin-7 (IL-7) can recruit lymphoid tissue-induced cells (LTi) to the lesion sites [14]. The surface of the LTi expressed the lymphotoxin-α1β2 ligand (LT-α1β2), which binds to its respective receptor LT-βR, expressed on the surface of stromal cells. After binding, it promotes stromal cells to secrete vascular endothelial growth factor C (VEGFC) that induces HEV formation and also facilitates the secretion of adhesion molecules used to recruit immune cells, such as the vascular cell adhesion molecule 1 (VCAM1) and the intercell adhesion molecule 1 (ICAM1) [15]. In an animal model of colorectal cancer, the process of TLS formation driven by bacteria-specific follicular helper T cells is similar to the process of the above [16]. 

In addition to the aforementioned LTi-dependent formation pathway of TLSs, CCL21 and CCL19 induce LTα1β2 expression in T cells, and CXCL13 stimulates LTα1β2 secretion expression in B cells, which both can recruit lymphocytes into TLSs via the LTβR signaling-dependent pathway from nearby HEVs [17,18]. Some other immune cells, such as macrophages, endothelial cells, immune fibroblasts, bone marrow mesenchymal stem cells (MSCs), and adipocytes, also play an important role in the formation and maturation process of TLSs [19]. Macrophages and endothelial cells secreted the cytokine IL-36γ, upregulated the expression of VCAM-1 and ICAM-1 in stromal cells and vascular endothelial cells, and promoted lymphocyte-recruitment capacity of HEVs through chemokines IL-8, CCL2, and CCL20, thus promoting TLS formation and maturation in human colorectal cancer [20,21]. Therefore, there may exist classical and non-classical TLS formation modes mentioned above in gastrointestinal tumors (Figure 1).

## 3. Recognition of the Characteristics of Tumor-Related TLSs in Gastrointestinal Tumors

In 1967, the infiltrating of nonspecific lymphocytes into tumor tissue was observed by the use of hematoxylin and eosin staining [22]. In 1987, lymphocyte aggregates with a “Crohn-like response” structure were found in the tumors of colorectal cancer patients by Graham et al. [23]. In 1992, as researchers studied the chronic inflammatory process, the concept of ‘tertiary’ lymphoid tissues was proposed due to the already existing presence of primary (including the bone marrow and thymus) and secondary lymphoid tissues (including lymph nodes, the spleen, tonsils, Peyer’s patches, and mucosa-associated lymphoid tissue [24]). Afterward, the description ‘tertiary lymphoid tissue’ was further used in a variety of diseases [2]. They have been observed in autoimmune diseases (Hashimoto’s thyroiditis, rheumatoid arthritis, myasthenia gravis, Sjogren’s syndrome, and multiple sclerosis), chronic microbial infection (hepatitis C, Helicobacter pylori, and Lyme disease), and chronic allograft rejection [3,4,5]. Soon, some researchers found similar structures in cancers. Structures similar to SLOs were found as TLS in melanoma [25] and non-small cell lung cancer (NSCLC) [26]. Afterward, TLSs were observed in gastrointestinal tumors [9,10,11].

With the development of pathological techniques and attention to the tumor microenvironment, the types of TLSs have gradually been revealed, which may vary between different cancer types and even in the different regions of same cancer. Currently, the maturity classification of TLSs is described as follows [27]: (i) aggregation (AGG): ambiguous lymphoid clusters; (ii) primary follicles (FL-I): circular lymphoid clusters without germinal centers; and (iii) secondary follicles (FL-II): follicles formed by germinal centers (GC). Usually, the AGG is considered the initial level of the TLSs. However, the FL-I and FL-II are considered to be the more mature TLSs. We can observe three different maturity levels of TLS in gastrointestinal tumors, and aggregations are the highest proportion of the three different maturity levels of TLSs in the gastrointestinal tumor [9,10,11,28,29]. The histological images of TLSs were showed in Figure 2.

To further recognize cellular compositions of TLSs, immunohistochemistry staining has been used to detect markers of specific lymphocytes, combined with computer quantitative image analysis to determine the expression of immune markers in the tumors, which further provide a theoretical basis for evaluating the relationship between TLS distribution, cellular components, and patient’s prognosis. In 2006, investigators began to analyze the cellular components of tumor-infiltrating lymphocytes (TILs), as well as the type, density, and location of immune cells in human colorectal tumors, to predict the clinical outcome [30]. Now, more and more research concentrates on the cellular components of TLSs. Yu et al. [31] found that CD4+ and CD8+ T cells were mainly distributed in the parafollicular cortex, with CD4+ T cells being more infiltrated than CD8+ T cells. Most CD20+ B cells were located in the follicular center, while CD11c+ DCs and CD45RO+ memory T cells were mainly located in the T cell region, partially dispersed into the follicular center. Only sporadic anti-NCR1+ NK cells were found, and CD68+ TAMs were rarely observed. FOXP3+ regulatory T cells (Tregs) were rare in both TLSs and tumor tissues. The distribution and composition of TLSs in gastric cancer [32,33] and colorectal cancer [29] are similar to those found in pancreatic neuroendocrine tumors by Yu. Park et al. [34] showed that the HEVs expressing characteristic peripheral node addressing protein (PNAd) and vascular addressing protein (MECA79) were located in the periphery of TLSs in gastric cancer (Table 1). 

Buisseret et al. [39] demonstrated that sections using H&E staining were less reproducible among pathologists, so gene array and single-cell profiling were used for detecting TLSs. For example, Coppola et al. [40] demonstrated the application of a 12-chemokine gene signature for TLS detection and identified unique ectopic lymph node-like structures in human primary colorectal cancer by immune gene array profiling. Jia et al. [41] detected the cell composition of TLSs by using single-cell profiling and proposed that natural killer T (NKT) cells mainly exist in gastric cancer tissues accompanied by mature TLSs. Due to the emergence of new technologies, we have an increasing number of means to evaluate TLSs in gastrointestinal tumors so that we can better identify TLSs in gastrointestinal tumors.

## 4. Anti-Tumor Mechanisms and Influencing Factors of the Tumor-Associated TLSs

### 4.1. Anti-Tumor Mechanisms of the Tumor-Associated TLSs

As previously described, the human immune system contains primary lymphoid organs, secondary lymphoid organs, and tertiary lymphoid organs (i.e., TLSs). Because they are anatomically similar to the SLOs, it has been suggested that the TLSs recapitulate the function of the SLOs in the TME [5]. Lymph nodes foster the encounter of antigen-laden APCs from tissues and naïve lymphocytes from the blood by providing a specialized niche that maximizes cell–cell contacts and thereby enables the generation of adaptive immune responses [42]. Compared with SLOs, the biggest difference in structures between TLSs and SLOs is that there is no membrane surrounding for TLSs, or a small amount of membrane [43], which provides a more convenient place for the movement of immune cells. Various active lymphocytes and immunoglobulin effector molecules in TLSs can more quickly move through HEVs or move freely into and out between lymphoid tissues and surrounding tissues [44] to exert the corresponding anti-tumor immune response. Meylan et al. [35] found that IgG-and IgA-producing plasma cells (PCs) spread along the tumor bed of fibroblasts, suggesting a mode of movement in which immune cells in TLS migrate to the surrounding tumor cell region in renal-cell carcinoma (RCC). This may exist the same way in gastrointestinal tumors because B cells in TLSs are associated with a favorable prognosis in gastric cancer [36].

TLSs are sites for lymphocyte induction and maturation, and they also provide a living place for follicular dendritic cells. FDCs are important antigen-presenting cells, which can present the exposed tumor-specific antigen (TSA) and tumor-associated antigen (TAA) to B cells and T cells in TLSs, driving them to become functional subsets. TLSs also provide a necessary place for effector T cells and effector B cells to exert anti-tumor immune responses [42]. When normal T and B cells mature, DCs timely present antigens to them and promote the immunoglobulin generation and effector T cell responses. In addition, B cells in TLS can also be used as antigen-presenting cells, which can present antigens to CD8+ T cells and further strengthen their immune responses [38]. As a result, the TLSs are considered as a local immune and anti-tumor micro battlefield, similar to a ‘local military academy’, providing a nearby location to fight against tumor cells. 

### 4.2. Influencing Factors of the Tumor-Associated TLSs

A few studies have indicated that TLSs have a negative impact on prognosis in patients with gastrointestinal tumors. As previously mentioned, the cellular components of TLSs include anti-tumor immune cells and also inhibitory immune cells. Schweiger et al. detected large numbers of regulatory T cells in patients with colorectal cancers, which might attenuate anti-tumor immune response [45]. Li et al. [46] proved that regulatory T cells were found in resected tumor samples of ovarian cancers and could significantly suppress the activation of CD8+ T cells in an IL-10- and TGF-β-dependent manner. The mechanism of immunosuppression by regulatory T cells in TLS of gastrointestinal tumors may be similar to this. Other immune inhibitory cells were also reported in TLSs of gastrointestinal tumors. Yamaguchi et al. [10] demonstrated that GATA3+ T helper 2 cells and CD68+ macrophages significantly increased in the recurrence group when assessing the influence of different subsets of TLSs in patients with curatively resected stage II/III colorectal cancers. In addition, Bento et al. found that HEVs of TLSs were rare in colorectal cancer but accumulate in extra-tumoral areas with disease progression and tumor cells may transfer via HEVs [47], which is different from the anti-tumor response of HEVs found in gastric cancer [34]. Therefore, TLS-associated Tregs and HEV presence may exert a negative influence on the capacity of TLSs to generate anti-tumor immune responses, although the detailed mechanism concerning how TLSs balance their immune responses based on these cellular components are still largely unknown. 

Apart from the inhibitory immune cells, such as regulatory T cells and HEVs in TLSs, Chen et al. [48] reported that the immune checkpoint molecules were also expressed in cells of TLSs. They found that the high expression of programmed death ligand 1 (PD-L1) or impaired human leukocyte antigen-I (HLA-I) expression in TLSs decreased the infiltration of B cells in the TME in esophageal adenocarcinoma. High expression of TIGIT on CD20 + B cells of TLSs was associated with poor prognosis in gastric cancer [49]. This may suggest that the upregulation of the immune checkpoint molecules in cells of TLSs can reverse their anti-tumor responses, so further studies to elucidate the expression patterns of immune checkpoint molecules in different cells of TLSs are urgently needed.

Interestingly, tumor metabolism not only has an impact on locally infiltrated immune cells but also may influence the anti-tumor immune responses of TLSs [50,51]. It was found that cancer cells need to take in far more glucose from their internal environment than normal cells to produce energy. This phenomenon is known as the “Warburg effect” [37]. Effector T cells have a selective requirement for glucose, while Tregs have a less glucose-dependent requirement. Therefore, tumor metabolism may be more favorable for Tregs. As previously mentioned, Tregs are an important suppressive cellular component of TLSs, and therefore tumor metabolism may lead to a decrease in the anti-tumor immunity of TLSs.

## 5. Prognostic and Predictive Potential of TLSs in Gastrointestinal Tumors

Based on the importance of TLSs in anti-tumor immune responses in gastrointestinal tumors, studies have also tried to assess their values in predicting the prognosis of patients with tumors. The favorable effects of TLSs on patients’ overall survival and recurrence-free survival have been observed in colorectal cancer [52], gastric cancer [53], and gastrointestinal stromal tumors [11]. Not only do the numbers of TLSs surrounding the tumors have an impact on patients’ outcomes, but also the cellular components of TLSs play a role in assessing patients’ prognosis.

### 5.1. Relationship between Cellular Components of TLSs and Prognosis

Based on the cellular components, Yamaguchi et al. [10] analyzed 353 TLSs found in 67 colorectal cases and divided TLSs into five types: germinal center-TLS type, B-cell-enriched-TLS type, FDC-rich-TLS type, Th-cell-enriched TLS type, and CTL/B/Th-TLS type. They found that auxiliary T cell-enriched TLSs were significantly associated with disease relapse in patients with advanced colorectal cancer, and Th2 cells were the major responsible cell subsets. Hong [34] proposed that high endothelial micro veins (HEVs) were the most important immune prognostic factors of RFS and OS in gastric cancer, while different results were found in colorectal cancer and liver cancer [27,47], and they found a negative impact of HEVs on patients’ prognosis. In gastric cancer, CD8 + T cells located in TLS are associated with improved prognosis [33]. As mentioned above, the presence of regulatory T cells in TLSs in patients with colorectal cancer may negatively affect the ability of TLSs to generate effector and memory T cells and was also proven to be negatively correlated with patients’ prognosis. Therefore, cellular components in TLSs influence their values in predicting the prognosis of patients, depending on the balance between anti-tumor subsets and inhibitory immune subsets.

### 5.2. Relationship between the Spatial Distribution of TLSs and the Prognosis

In a study of TLSs in hepatocellular carcinoma, Julien Calderaro et al. [54] found that TLSs within the tumor were associated with a low risk of early recurrence (less than two years after surgery), and they had no correlation with the risk of late (more than two years) recurrence. In contrast, TLSs in tumor-adjacent liver tissue had no prognostic value for early and late recurrence, which highlighted the prognostic impact of TLS location in the prediction of patients’ outcomes. TLSs located in the non-tumor liver only contributed to local inflammation, while TLSs within the tumor core region truly reflect effective anti-tumor immunity. Ding et al. [55] also showed that the spatial distribution and abundance of TLSs were significantly correlated with patient’s prognosis in intrahepatic cholangiocarcinoma. Sofopoulos et al. [56] defined the TLS around breast tumors as an area within 5 mm from the invasive margin and subdivided the TLS into adjacent and distal TLS based on the distance and interval between the normal breast tissue and the invasive margin. Adjacent TLSs are thought to be TLSs located on the surface of the invasive margin, and the distal TLSs are thought to be a class of TLSs in which normal breast tissue is inserted between it and the invasive margin of the tumor. The proximal high density of TLSs was associated with reduced disease-free survival (DFS) but not overall survival (OS), and distal TLS density was negatively associated with both patients’ DFS and OS. Almost no study revealed the prognostic significance of TLS location in gastrointestinal tumors. Most of the TLSs with good prognoses in gastrointestinal tumors are located at the edge of tumor invasion and in the tumor stroma. Only Wang et al. found that TLSs at the edge of tumor invasion are positively related to prognosis, while TLS within the tumor is not related to prognosis [29]. All these results suggested the different locations of TLSs in gastrointestinal tumors may associate with different prognoses.

### 5.3. Prognostic Role of TLSs in Combination with Other Indicators

The prognostic factors of tumor patients are multifaceted, and a single TLS index sometimes does not accurately predict the prognosis of patients. In recent years, there has been more and more research on the prediction model of the combined nomogram. Yu et al. [31] identified TNM stage, WHO grade, and TLS as independent prognostic factors and combined these factors to draw a nomogram, which greatly improved the predictive power of the prediction model. Similarly, Yamakoshi et al. found that the combination of neutrophil-to-lymphocyte ratio (NLR) and TLSs were useful for the stratification of patient prognosis of gastric cancer [57]. Wang et al. [29] also proposed that combining TLSs and tumor stroma percentage (TSP) can better predict the prognosis of patients with non-metastatic colorectal cancer. All these reveal that the combination of TLSs and other indicators may be useful for the stratification of patient prognosis in gastrointestinal tumors.

### 5.4. Prediction Values of TLSs in Immune Therapy

At present, adjuvant therapies for tumors include radiotherapy, chemotherapy, targeted therapy, and immunotherapy, which all have an impact on immune cells in the TME. In recent years, immune checkpoint blocking (ICB) treatment is getting hotter and hotter, but the effectiveness of ICB treatment in gastrointestinal tumors is not totally satisfying and still needs improvement [58,59,60,61,62,63,64,65,66,67,68,69,70,71] (Table 2). It is urgent to find an index that can predict the response of immunotherapy. One recent study [28] reported that 13 patients with gastric cancer had a higher TLS score after receiving programmed cell death 1 (PD1) blockade therapy, suggesting that the generation of TLSs could occur after immune therapy which was also shown to be a biomarker for good responsiveness of immune therapy. Similar conclusions have been found in tumors of other systems. For instance, Solinas et al. [72] found that the high level of TILs and TLSs were correlated with the high expression of immune checkpoints in human breast cancer, and proposed TLSs may predict the effectiveness of immunotherapy. Helmink et al. demonstrated that the presence of B cells and TLSs can make predictions for the good effects of ICB treatment in patients with melanoma and renal cell carcinoma [73]. Therefore, we have reason to believe that TLSs may be a good indicator to predict the effectiveness of immunotherapy for gastrointestinal tumors.

## 6. Therapeutic Strategies for Inducing TLS Formation in Gastrointestinal Tumors

The favorable correlation of TLSs with gastrointestinal tumors has led the investigators to propose one innovative immunotherapy strategy for inducing TLS formation in the TME. DG et al. [74] observed newly formed tumor-associated TLSs in a preclinical mouse model (gp130F/F) of gastric cancer, where tumorigenesis is dependent on hyperactive STAT3 signaling through the common IL-6 family signaling receptor, gp130. This may be a possible way to induce TLSs formation in immunotherapy of gastric cancer. Weinstein et al. [75] found that direct injection of dendritic cells (DC, expressing T-cell-specific transcription factor T-bet) into a mouse colorectal cancer xenograft model promoted lymphocyte infiltration and TLSs generation and slowed tumor growth. This antitumor effect is dependent on IL-36γ and can be blocked by IL-36 receptor antagonists or IL-36 receptor defects. Chimeric antigen receptor T (CAR-T) cells infusion is an important method in the therapy of some solid tumors. Tamada [76] has developed IL-7 and CCL19-expressing CAR-T cells to form T cell zone-like structures (i.e., TLSs) in tumor tissues. 

Furthermore, it has been reported that immune checkpoint inhibitors (ICIs) can induce TLSs formation with an anti-tumor function in the TME [77], such as a post-treatment examination of tumor resections from 20 patients with NSCLC who were treated in phase II clinical trial of neoadjuvant nivolumab (anti-PD-1), which showed the occurrence of TLSs, while in the specimens of non-responsive patients’ TLSs de novo were either absent or rare [78]. 

In addition, evidence that TLSs can be therapeutically induced by tumor vaccine, for instance, in patients with high-grade cervical intraepithelial neoplasia (CIN2/3), is observed, where TLS formation and clonal expansion of TLSs could be observed in regressing lesions after vaccination against the human papillomavirus oncoproteins [79]. Similarly, one latest clinical result also supports this treatment strategy by recruiting immune cells to the TME and the formation of TLSs. Among 39 patients with pancreatic ductal adenocarcinomas (PDACs), 33 of 39 patients with excised PDAC tissue showed vaccine-induced TLS occurrence. Immunotherapy-inducing TLS generation turned a “non-immunogenic” tumor into an “immunogenic” tumor after two weeks of treatment, further proving that immunotherapy can orchestrate infiltrating T cells to form TLSs, which in turn contributes to the formation of the anti-tumor microenvironment [80]. Other strategies have utilized biomaterials to induce TLSs. These materials can support the formation of TLSs by locally and controllably releasing chemokines and providing cellular support. In one study, collagen sponge scaffolds embedded with sustained-release gel beads containing LT-α1β2 and many chemokines were transplanted into the subcapsular space of mice to establish TLSs, recruiting memory T cells and B cells to induce a strong antigen-specific immune response [81]. 

Lastly, neoadjuvant chemotherapy can also lead to the regeneration of TLSs. For example, a study reported the presence of numerous intra-tumoral TLSs and antigen-presenting cells (DC-LAMP) in 11 hepatoblastoma patients receiving cisplatin-based neoadjuvant chemotherapy [82]. 

Therefore, associated signaling pathways, therapeutic cells, immune checkpoint inhibitors, tumor vaccines, biomaterials, and neoadjuvant chemotherapy may all drive the generation of TLSs, which led to a better anti-tumor immune response in the therapy of gastrointestinal tumors.

## 7. Conclusions and Future Perspectives

In conclusion, gastrointestinal tumor-related TLSs exist in the local area of tumor regions and are an important part of anti-tumor immunity in TME. The presence of TLSs is usually an indicator of a good prognosis for gastrointestinal tumors. Further studies on exploring the existence of TLSs in other types of gastrointestinal tumors, such as gastric neuroendocrine neoplasms (G-NENs), which are usually considered as ‘immune cold tumors’, will help to better understand the role of TLSs in tumor immunity. In addition, with an increased and effective application of adjuvant therapy and immunotherapy, it is also urgently needed to elucidate whether TLSs are the major local immune response sites or even the only existing regions for the generation and maturation of effector immune cells, especially under the condition that the number of lymph nodes may decrease or even disappear after adjuvant therapy. Based on the complex cellular components in TLSs, it is also crucial to further address how to exert proper anti-tumor immune responses of TLSs in the presence of inhibitory immune subsets in TLSs. Therapeutic strategies targeting TLSs have provided promising results in animal models, which also need to be translated into clinical trials in the future.

## Figures and Tables

**Figure 1 cancers-15-00367-f001:**
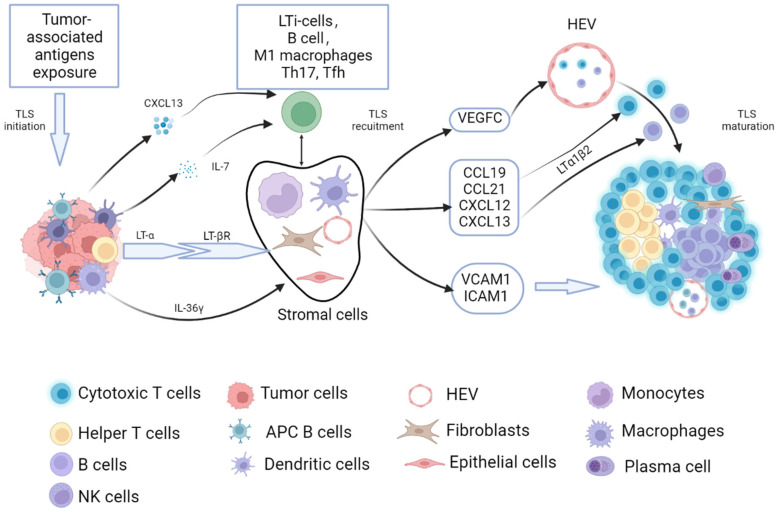
The process of tumor-related TLSs formation and maturation. The process of tumor-related TLS formation and maturation have similarities with that of SLOs. When the tumor-associated antigens exposed to the TME caused an anti-tumor immune response, adaptive immune cells were activated, and then secreted lymphotoxin-α (LT-α) enabled them to interact with the corresponding receptor (lymphotoxin-β receptor, LT-βR) expressed on stromal cells (i.e., endothelial cells, fibroblasts, epithelial cells, monocytes, and dendritic cells). The secretion of CXCL13 and IL-7 can recruit the LTi to the lesion sites. The surface of the LTi expressed LT-α1β2, which can bind to LT-βR. After the binding, it promotes stromal cells to secrete vascular endothelial growth factor C (VEGFC)-induced HEV formation. It can also facilitate the secretion of adhesion molecules used to recruit immune cells, such as the vascular cell adhesion molecule 1 (VCAM1) and the intercell adhesion molecule 1 (ICAM1). CCL21 and CCL19 induce LTα1β2 expression in T cells, CXCL13 stimulates LTα1β2 expression in B cells, and it can recruit lymphocytes into TLS via the LTβR signaling-dependent pathway from nearby HEVs. Besides, the cytokine IL-36γ secreted by macrophages and endothelial cells contributes to TLS formation and maturation in gastrointestinal tumors.

**Figure 2 cancers-15-00367-f002:**
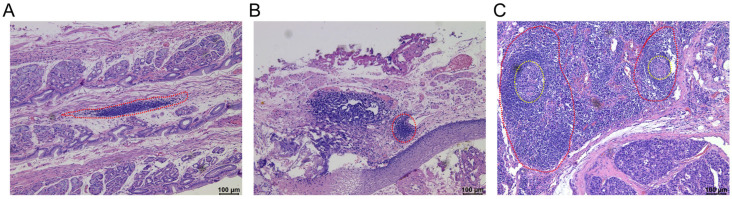
The maturity classification of tertiary lymphoid structures. Representative images of tertiary lymphoid structures (TLSs) detected in formalin-fixed paraffin-embedded tumor sections by haematoxylin and eosin (H&E) staining. (**A**) Aggregation (AGG) (The region of red dotted line); (**B**) Primary follicles (FL-I) (The region of red dotted line); (**C**) Secondary follicles (FL-II) (The region of red dotted line with a germinal center) (The region of the yellow dotted line represents the germinal center). These histological images are original, unpublished images from the authors’ examination of tumors with gastric neuroendocrine neoplasms.

**Table 1 cancers-15-00367-t001:** Characteristics of tumor-related TLS in different gastrointestinal tumors.

		Characteristic Detecting Markers	Subpopulation Markers	Prognoses	Reference
Gastric cancer	B cell zone	CD20^+^ B cells		Favorable	[9,32,33,35,36,37]
			GC B cells (Bcl-6^+^ CD20^+^)	Favorable	[35]
			FDCs (CD21^+^)	No evaluation	[32,33,35]
			Macrophage (CD68^+^)	Unfavorable	[37]
	T cell zone	CD3^+^ T cells		Favorable	[32,35,37]
			Help T cells (CD4^+^)	No evaluation	[33,35]
			Th1 (CD4^+^ T-bet^+^)	Favorable	[9]
			Tregs (CD4^+^ FOXP3^+^)	Unfavorable	[9]
			Cytotoxic T cells (CD8^+^)	Favorable	[32,35,37]
			Memory T cells (CD45RO^+^)	No evaluation	
	HEVs	PNAd or MECA79	PNAd or MECA79	Favorable	[33,35]
Colorectal cancer	B cell zone	CD20^+^ B cells	Tfh cells (CD3^+^ CD8^-^ Bcl-6^+^)	No significance	[10]
			FDCs (FDC^+^)	No significance	[10,29]
			GC B cells (Bcl-6^+^ CD20^+^)	No significance	[10]
			Macrophage (CD68^+^)	Unfavorable	[10,29]
	T cell zone	CD3^+^ T cells	CTL (CD8^+^)	No significance	[10]
			Th1 (CD3^+^ T-bet^+^)	No significance	[10]
			Th2 (CD3^+^ GATA3^+^)	Unfavorable	[10]
			Th17(CD3^+^ ROR-γT^+^)	No significance	[10]
			Tregs (CD3^+^ FOXP3^+^)	Unfavorable	[10,38]
			Memory T cells (CD45RO^+^)	No evaluation	[29]
	Other immune cells	NCR1^+^ NK cells		No significance	[29]
		CD15^+^ TAN cells		No significance	[29]
	HEVs	PNAd or MECA79	PNAd or MECA79	Unfavorable	[27]
Gastrointestinal Stromal Tumors (GIST)	B cell zone	CD20^+^ B cells	Naive B cells (Bn) (CD20^+^ CD27^−^ IgM^+^)	No evaluation	[11]
			IgM^+^ memory B cells (IgM^+^ Bm) (CD20^+^ CD27^+^ IgM^+^)	No evaluation	[11]
			CD27^−^ isotypeswitched memory B cells (CD27^−^ Sw Bm) (CD20^+^ CD27^−^ IgM^−^)	No evaluation	[11]
			CD27^+^ isotype-switched memory B cells (CD27^+^ Sw Bm) (CD20^+^ CD27^+^ IgM^−^)	No evaluation	[11]
			Plasma cells (PCs) (CD20-CD24−CD27hiCD38hi)	No evaluation	[11]
	T cell zone	CD3^+^ T cells	CTL (CD8^+^)	No significance	[11]
			Th1 (CD4^+^ T-bet^+^)	No significance	[11]
			Th2 (CD4^+^ GATA3^+^)	No significance	[11]
			Th17(CD4^+^ ROR-γT^+^)	No significance	[11]
			Tregs: (CD4^+^ FOXP3^+^)	Unfavorable	[11]
	Other immune cells	Tissue-resident memory T cells	CD103^+^ Trm	No evaluation	[11]
	HEVs	PNAd or MECA79	PNAd or MECA79	No detected	

**Table 2 cancers-15-00367-t002:** The main outcomes of immune checkpoint blocking treatment in different gastrointestinal tumors.

	Tumor Type	Medication Plan	Key Outcome	Effectiveness	Reference
Gastric cancer and Gastroesophageal junction cancer	GC	Pembrolizumab + lenvatinib	ORR 69%	Positive	[68]
	GC/GEJC	Pembrolizumab + SOX	ORR 72.2%	Positive	[58]
	GEJC	Margetuximab + pembrolizumab	ORR 18%	No significance	[67]
	GC/GEJC	Pembrolizumab	ORR 25.8%	No significance	[64]
	GC/GEJC	Pembrolizumab	ORR 14.5%	No significance	[63]
	GC/GEJC	Nivolumab + CapeOX	ORR 76.5%	Positive	[59]
Colorectal cancer	MSI-H mCRC	Pembrolizumab	ORR 54%	Positive	[65]
	MSS mCRC	Pembrolizumab	ORR 0%	Failed	[65]
	Advanced CRC	Nivolumab	ORR 32%	No significance	[66,70]
	Advanced CRC	Nivolumab+ Ipilimumab	ORR 55%	Positive	[66,70]
	MSS CRC	Regorafenib+ Nivolumab	ORR 36%	No significance	[60]
Gastrointestinal Stromal Tumors (GIST)	Advanced GIST	Nivolumab+Ipilimumab	CR 6.7%	No significance	[61]
	Advanced GIST	Dasatinib+Ipilimumab	ORR 53.8%	Positive	[71]
Gastrointestinal neuroendocrine neoplasms (GE-NEN)	Recurrent or Metastatic GNENs	Toripalimab	ORR 20%	No significance	[69]
	NENs (GNENs47%)	Ipilimumab +nivolumab	ORR 25%	No significance	[62]

ORR: objective response rate. CR: complete response. MSI—H: microsatellite instability—high. mCRC: metastatic colorectal cancer. MSS: microsatellite stability. CRC: colorectal cancer.

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
