# Peer review of "Turning Tertiary Lymphoid Structures (TLS) into Hot Spots: Values of TLS in Gastrointestinal Tumors"

_cancers, 2023, doi:10.3390/cancers15020367_

Round 1
Reviewer 1 Report
This review focuses on the role of TLSs in gastrointestinal tumors, summarizing the mechanisms of their anti-tumor immune responses in an exhaustive and coherent way.
1)page 7 line 294, the authors state that…. “In recent years, immune checkpoint blocking treatment is getting hotter and hotter, but the effectiveness of ICB treatment is not high”. Please cite the main studies of immunotherapy in gastrointestinal tumors, specifying which were positive and which failed, possibly integrating the manuscript with a table. 2)page 7 line 301: the sentence “Solinas et al [58] found that the high expression level of immune checkpoint molecules was correlated with the high formation level of TILs and TLSs which may predict the ….” is not easily understood, please rephrase to make it clearer and more understandable
Author Response
Point 1: Page 7 line 294, the authors state that…. “In recent years, immune checkpoint blocking treatment is getting hotter and hotter, but the effectiveness of ICB treatment is not high”. Please cite the main studies of immunotherapy in gastrointestinal tumors, specifying which were positive and which failed, possibly integrating the manuscript with a table.
Response 1: We thank the Reviewer for this important point. We have listed the main studies of immunotherapy in gastrointestinal tumors in Table2(Page12 Line 406). In the table, we listed the key message of these clinical trials which contained positive and failed. We can conclude that the effectiveness of ICB treatment is not very high.
Point 2: Page 7 line 301: the sentence “Solinas et al [58] found that the high expression level of immune checkpoint molecules was correlated with the high formation level of TILs and TLSs which may predict the ….” is not easily understood, please rephrase to make it clearer and more understandable.
Response 2: Thanks for your comment. We have rewritten the sentences and make it clearer and more understandable(Page7 Line 305-308).
Reviewer 2 Report
The authors did an excellent job of reviewing the clinical significance and biology of the tertiary lymphoid structure in gastrointestinal tumors.
This paper organized the recently discovered relationship between TLS and prognosis by dividing them into specific sectors -e.g., in combination with other indicators and spatial distribution. This is a strength of this paper.
However, it is a bit lengthy and could be shortened for readability. Sentences in lines 129 - 136 may be unnecessary.
Author Response
Point 1: This paper organized the recently discovered relationship between TLS and prognosis by dividing them into specific sectors -e.g., in combination with other indicators and spatial distribution. This is a strength of this paper. However, it is a bit lengthy and could be shortened for readability. Sentences in lines 129 - 136 may be unnecessary.
Response 1: Thank you very much for your suggestion and we have deleted the corresponding contents(Page 3 Line 133).
Reviewer 3 Report
The paper "Turning tertiary lymphoid structures (TLS) into hot spots : values of TLS in gastrointestinal tumors" is a very well written and well structured review and personally I don't have any observations to make except one that despite being a review does not histological images has been inserted in the paper that can make the histological characteristics and the precise localization of the TLS clear to the reader, therefore it is important to add an explanatory iconographic part before publication
Author Response
Point 1: The paper "Turning tertiary lymphoid structures (TLS) into hot spots : values of TLS in gastrointestinal tumors" is a very well written and well structured review and personally I don't have any observations to make except one that despite being a review does not histological images has been inserted in the paper that can make the histological characteristics and the precise localization of the TLS clear to the reader, therefore it is important to add an explanatory iconographic part before publication.
Response 1: Thank you very much for your constructive suggestions. Representative images of TLSs (Figure 2), Aggregation (AGG) (Figure 2A), FL-I (Figure 2B), and FL-II (Figure 2C) were added in the latest version of manuscript(Page10 Line399-403).